

# Exploring the scale effect of nonpoint source pollution risk on water quality in Lake Basins of Central Yunnan Plateau using the Minimum Cumulative Resistance model

Li Fu[1,2,*], Xiaoliang Ma[3,*], Shuangyun Peng[1,2], Luping Gong[1,2], Rui Zhang[1,2] and Bangmei Huang[4]

[1] Faculty of Geography, Yunnan Normal University, Kunming, China
[2] Center for Geospatial Information Engineering and Technology of Yunnan Province, Kunming, China
[3] School of Ecology and Environmental Science, Yunnan University, Kunming, China
[4] Kunming No.10 High School, Kunming, China
[*] These authors contributed equally to this work.

Corresponding author
Shuangyun Peng, frankmei@126.com

## ABSTRACT

Nonpoint source (NPS) pollution has emerged as the predominant water environment issue confronting plateau lakes in central Yunnan. Quantitative analysis of the impact of NPS pollution on water quality constitutes the key to preventing and controlling water pollution. However, currently, there is a dearth of research on identifying NPS pollution risks and exploring their relationship with water quality based on the Minimum Cumulative Resistance (MCR) model in the plateau lake basins of central Yunnan. Particularly, studies on the spatial heterogeneity of the impact of NPS pollution on water quality from a multi-scale perspective are scarce. Therefore, this study focuses on three typical lake basins in the Central Yunnan Plateau–Fuxian Lake, Xingyun Lake, and Qilu Lake (the Three Lakes). Utilizing the MCR model to identify NPS pollution risks, the study analyzes seven different scales, including sub-basins, riparian buffer zones (100 m, 300 m, 500 m, 700 m, and 1,000 m) and lakeshore zones, to reveal the multi-scale effects of NPS pollution on water quality through correlation analysis. The results indicate that: (1) Over 60% of the areas in the Three Lakes Basin are at high or extremely high risk, mainly concentrated in flat terrain and around inflow rivers; (2) The area of NPS pollution from paddy field source landscape (PFSL) is greater than that from construction land source landscape (CLSL), and the high-risk areas of NPS pollution are also larger for PFSL compared to CLSL; (3) The mean resistance values of PFSL and CLSL show a significant negative correlation with monthly mean values of water quality indexes ($NH_3$-N, TP, $COD_{Cr}$), with the 1,000 m riparian buffer zone scale showing the greatest correlation with most water quality indexes, especially $NH_3$-N; (4) The correlation between the mean resistance value of CLSL and the monthly mean values of water quality indexes is significantly higher than that of PFSL, indicating a greater impact of CLSL on water quality compared to PFSL. In summary, PFSL and CLSL are the primary sources of NPS pollution in the Three Lakes Basins. The 1,000 m riparian buffer zone scale is the most sensitive to the impact of NPS pollution on water quality. This study provides scientific references for landscape pattern optimization and

precise control of NPS pollution risks in the Central Yunnan Plateau lake basins and offers a new research perspective for exploring multi-scale effects of NPS pollution on water quality.

## INTRODUCTION

Plateau lakes are vital resources for social and economic development, essential for human survival, and key factors affecting regional ecological environments (*Zhang et al., 2019*). As socioeconomic conditions improve, intensified urban expansion and the interim exploitation of surface resources have compromised the water quality and ecological equilibrium of plateau lakes, resulting in significant degradation (*Van Vliet, Floerke & Wada, 2017*). In the current water pollution control and environmental protection work, the impact of point source pollution on water quality has been effectively controlled (*Yang et al., 2019b*; *Zhao et al., 2024*). However, nonpoint source (NPS) pollution often occurs in many situations such as fertilizer use in agricultural production, urban rainwater runoff, and animal husbandry. Due to the dispersion and uncertainty of its emissions, it is difficult to achieve effective centralized control, which is one of the main causes of water quality deterioration (*Chen et al., 2018*; *Longyang, 2019*; *Pan et al., 2023*; *Qian et al., 2020*; *Zhao et al., 2023*). Identifying NPS pollution risk areas and exploring their multi-scale impacts on water quality are of great significance for water environment governance, ecological environmental protection, and ecological security construction in plateau lake basins.

NPS pollution seriously affects agricultural production (*Cristan et al., 2016*; *Wang et al., 2019a*), water resources, and aquatic habitats due to its randomness, universality, hysteresis, fuzziness, and latency (*Kumwimba et al., 2018*; *Wang et al., 2016a*). Effective identification of NPS pollution risk zones is vital for plateau lake water quality management (*Edwards & Withers, 2008*; *Longyang, 2019*). Identification methods mainly include field investigation method and model identification method (*Guo, Wang & Zhu, 2004*; *Jiang et al., 2014*). While field surveys have high accuracy, they are resource-intensive and less universal (*Lu et al., 2011*; *Zhu, Schmidt & Bryant, 2012*; *Zhu et al., 2011*). Model identification methods are more accessible and simulate ecological processes involved in NPS pollution (*Giri, Nejadhashemi & Woznicki, 2012*; *Jiang et al., 2014*; *Liu et al., 2016*).

The model method is simple, efficient, and can accurately identify the risk status of nitrogen and phosphorus pollution in the study area. Models for NPS pollution risk identification include the Soil Water and Assessment Tool (SWAT) (*Krysanova & Srinivasan, 2015*), nitrogen (NI) phosphorus (PI) index (*Bechmann et al., 2009*), and minimum cumulative resistance (MCR) model (*Wang et al., 2016b*). The SWAT model accurately identifies NPS pollution risk zones (*Abbaspour et al., 2007*; *Gassman et al., 2007*; *Krysanova & Srinivasan, 2015*) and has been widely used (*Bechmann et al., 2009*; *Ding et*

*al., 2020*; *Giri, Nejadhashemi & Woznicki, 2012*; *Liu et al., 2016*; *Liu et al., 2013*; *Meng & Wang, 2017*; *Ouyang et al., 2016*; *Panagopoulos, Makropoulos & Mimikou, 2011*; *Singh et al., 2012*; *Wang et al., 2016a*; *Xu et al., 2016*). The NI-PI index model identifies nitrogen and phosphorus pollution risk zones (*Birr & Mulla, 2001*; *Hughes, Magette & Kurz, 2005*; *Shen et al., 2011*) and has been applied by scholars (*Ping et al., 2011*; *Zhou & Gao, 2008*; *Zhou & Gao, 2011*).

However, the SWAT model is complex, data-intensive, and has low calculation efficiency, while the NI-PI index model is subjective and does not fully explain actual pollution loads. In contrast, the MCR model, based on the "source–sink" theory, calculates the resistance encountered by NPS pollutants moving from the "source" to any point in space (*Dong et al., 2022*; *Li et al., 2015*; *Wang et al., 2023*; *Wang et al., 2016b*; *Xue et al., 2022*). The advantages of the model are as follows: Firstly, it features convenient data processing, a simple analysis method, and provides intuitive results. Additionally, it takes into account the influence of distance and resistance between units, as well as quantitatively reflects the accessibility of NPS pollutants to water bodies (*Dai, Liu & Luo, 2021*; *Jin et al., 2020*; *Kong et al., 2018*; *Wang et al., 2016b*; *Wei et al., 2022*; *Xu et al., 2021*). Secondly, based on the characteristics of the study area, the model selects corresponding impact factors to assess the risk of NPS pollution according to local conditions, yielding results that closely align with actual pollution in the area (*Fu, Zhang & Wang, 2024*). Thirdly, considering spatial heterogeneity, the model can identify and analyze NPS pollution at different spatial scales (*Wu, 2004*; *Wu, 2007*). It simplifies basin complexity into multi-scale analysis units which helps reveal differences in impact across multiple scales. The simulation results are more consistent with real geographical phenomena. The model has been applied in large-scale opencast coal mine areas, Three Gorges reservoir area of Yangtze River and Haihe River Basin for identifying NPS pollution risks with research outcomes aligning with actual situations (*Kong et al., 2018*; *Wang et al., 2016b*; *Xu et al., 2021*).

NPS pollution poses a significant challenge in high-altitude lake basins. Current research on NPS pollution identification using the MCR model and its relationship with water quality has limitations: (1) focus on the sub-basin scale (*Kong et al., 2018*), neglecting the scale effect of NPS pollution on water quality; (2) emphasis on plain and hilly zones (*Kong et al., 2018*; *Wang et al., 2016b*), with less attention to the unique plateau lake basins in central Yunnan. Fuxian Lake, Xingyun Lake, and Qilu Lake are important freshwater lakes in the central Yunnan Plateau, characterized by low latitude and high-altitude features. They are crucial for studying biodiversity formation mechanisms and are sensitive to global change, affected by the East Asian and Southwest Monsoons. These lakes attract international research (*Apudo et al., 2016*; *Chen et al., 2019*; *Du et al., 2019*; *Wu et al., 2016a*) and support the development of the Central Yunnan Economic Zone and Yuxi city (*Yang et al., 2019a*). However, rapid economic growth and population expansion seriously threaten water quality due to NPS pollution (*Gao et al., 2015*), necessitating restoration and protection of the basins' water ecological security.

This study takes the typical plateau lake basins in Yunnan Province (Fuxian Lake, Xingyun Lake, and Qilu Lake) as research cases. Based on the MCR model, the multi-scale response relationship between water quality and NPS pollution risk is discussed under the

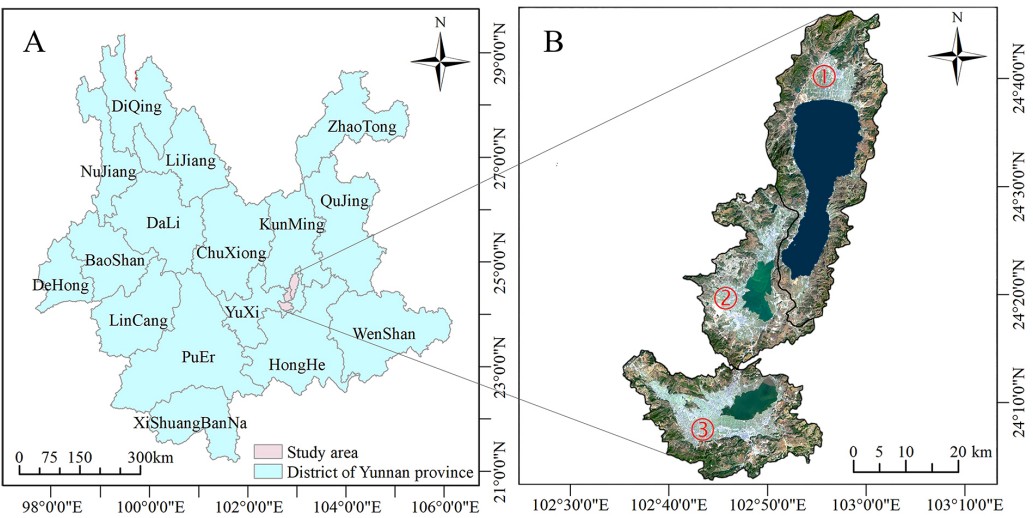

**Figure 1** **Overview of the study area.** (A) The location of the study area. (B) The remote sensing image of the study area. Marks ① to ③ indicate Fuxian Lake, Xingyun Lake, Qilu Lake, respectively.

multi-scale conditions of sub-basins, riparian buffer zones (100 m, 300 m, 500 m, 700 m, 1,000 m), and lakeshore zones. The objectives of this article are as follows: (1) Compare the effect of NPS pollution on water quality at multiple scales; (2) Reveal the spatial scale effects and differences of the impact of NPS pollution on the water quality of rivers entering the lakes; (3) Provide decision-making basis for scientific prevention and control of NPS pollution in the lake basin of the central Yunnan Plateau.

## MATERIALS AND METHODS

### Study area

Fuxian Lake, Xingyun Lake, and Qilu Lake (the Three Lakes) are important plateau lakes in central Yunnan, belonging to the Nanpan River system in the Pearl River Basin. The Three Lakes Basin, located on the western margin of the Yunnan-Guizhou Plateau, with a terrain of north, middle and low (Fig. 1). Fuxian Lake Basin (675.32 km²) is a deep-water, nutrient-poor lake with excellent water quality, despite some inflow rivers being Class V or inferior due to human activities. Xingyun Lake Basin (406.815 km²) is seriously affected by NPS pollution from the phosphorus industry, with water quality generally classified as Class V and inferior Class V from 2003 to 2016 (*Luo et al., 2022*). Qilu Lake Basin (354.99 km²) has inflow rivers remaining at levels V and inferior V, indicating serious deterioration (*Klamt et al., 2021*). In general, the Three Lakes Basin are affected by urbanization, farmland activities, industrial and agricultural production, domestic sewage and other factors, and NPS pollution is becoming increasingly prominent, and the water quality of the basins is seriously damaged.

### Data

The data types of this study mainly include: (1) Digital Elevation Model (DEM), we also use DEM data to extract the boundaries, slope, and surface roughness information of

**Table 1 Remote sensing image data descriptions.**

| Sensor | Date | GTM | Scene ID | Spatial resolution (m) |
|---|---|---|---|---|
| OLI | 2018-03-01 | 03:34:41 | LC81290432018060LGN00 | 30 |

**Table 2 Three Lakes Basin Data.**

| Data type | Year | Spatial resolution | Data source |
|---|---|---|---|
| Terrain | | | https://glovis.usgs.gov/ |
| Land use type | 2018 | 30 m | http://www.resdc.cn |
| Vegetation coverage | | | http://www.dsac.cn/ |
| Water quality | | – | Yuxi Environmental Protection Bureau |

the Three Lakes basin; (2) Land use data of the Three Lakes basin in 2018; (3) Landsat-8 OLI remote sensing image data in 2018, we use this data to extract the basin vegetation coverage data (Table 1); (4) Water quality data of rivers entering the lake. Since some rivers were interrupted in flow in 2018. This article calculated the monthly mean water quality sampling point data of each river entering the lake in 12 months of 2018, and selected the main water quality indexes $NH_3$-N, TP, TN, $COD_{Mn}$, $BOD_5$, $COD_{Cr}$ and fluoride for analysis. The sources of the above data and their resolution information are detailed in Tables 2 and 3.

## Methods
### Minimal cumulative resistance model
The Minimum Cumulative Resistance (MCR) model is utilized for assessing the resistance encountered by species diffusion or ecological flow in space in ecological processes (*Fu, Zhang & Wang, 2024*; *He et al., 2024*). MCR was first proposed by *Knaapen, Scheffer & Harms (1992)*, and it reflects the MCR value calculated by overcoming resistance in the process of starting from the "source" and passing through different resistance landscape units and generates the MCR surface to characterize the risk of nonpoint source pollution in the study area. The model considers three factors: source, space distance, and resistance base surface. The expression is as follows:

$$MCR = f_{min} \sum_{j=n}^{i=m} (D_{ij} g R_i) \tag{1}$$

where MCR is the minimum cumulative resistance value; $f$ is an unknown monotone increasing function; $D_{ij}$ is the spatial distance from nonpoint source pollution source j to landscape unit i; and $R_i$ is the resistance coefficient of landscape unit i to the movement process.

The MCR model can directly reflect the resistance to pollutant migration in different areas of the landscape. In this article, the MCR model is used to quantify the resistance value of each unit, so as to identify the high risk areas of NPS pollution (*Teng & Zhou,*

**Table 3** Monthly mean value of water quality indexes monitoring data of main inflow rivers in the Three Lakes Basin (mg/L).

| Water quality index/River | NH$_3$-N | TP | TN | COD$_{Mn}$ | BOD$_5$ | COD$_{Cr}$ | Fluoride |
|---|---|---|---|---|---|---|---|
| Daicun River | 0.13 | 0.15 | 4.58 | 2.85 | 1.50 | 11.00 | 0.75 |
| Dongda River | 0.33 | 0.16 | 4.47 | 3.46 | 2.61 | 13.05 | 0.39 |
| Fucheng River | 1.74 | 0.40 | 4.33 | 4.55 | 4.58 | 14.82 | 0.50 |
| Ge River | 0.15 | 0.04 | 0.68 | 2.20 | 2.83 | 12.01 | 0.28 |
| Jianshan River | 0.11 | 0.07 | 0.76 | 2.73 | 2.18 | 8.73 | 0.36 |
| Liangwang River | 0.13 | 0.09 | 3.57 | 2.18 | 2.25 | 11.03 | 0.20 |
| Lujv River | 1.02 | 0.24 | 9.55 | 4.97 | 5.00 | 17.43 | 0.34 |
| Niumo River | 0.78 | 0.09 | 6.99 | 4.40 | 2.75 | 16.05 | 0.45 |
| Shanchong River | 0.23 | 0.13 | 2.32 | 3.25 | 3.10 | 13.10 | 0.47 |
| Dajie River | 4.89 | 1.44 | 8.28 | 8.98 | 15.49 | 47.50 | 0.56 |
| Dalongtan River | 0.67 | 0.13 | 2.41 | 3.45 | 3.43 | 15.92 | 0.18 |
| Dazhuang River | 3.64 | 0.48 | 7.71 | 4.78 | 8.37 | 28.58 | 0.39 |
| Dongxida River | 2.27 | 0.61 | 10.01 | 4.23 | 5.77 | 21.75 | 0.36 |
| Luosipu River | 1.67 | 0.55 | 13.13 | 3.26 | 4.23 | 19.42 | 0.65 |
| Xue River | 1.03 | 0.24 | 3.58 | 2.71 | 3.51 | 15.17 | 0.24 |
| Yucun River | 2.25 | 0.29 | 4.42 | 8.24 | 7.17 | 38.73 | 0.50 |
| Zhoudeying River | 0.59 | 0.23 | 1.36 | 2.51 | 3.23 | 14.75 | 0.25 |
| Hongqi River | 0.98 | 1.01 | 9.69 | 8.94 | 8.25 | 42.00 | 0.93 |
| Zhong River | 3.59 | 0.80 | 10.52 | 9.84 | 12.67 | 43.25 | 0.75 |

*2023*; *Wu, Pan & Zhu, 2024*). Compared with other methods, it shows unique advantages in dealing with landscape resistance, predicting pollutant paths, and identifying risk areas (*Ye et al., 2015*).

**Pearson correlation coefficient**

Pearson correlation coefficient is a statistical analysis method widely used in various disciplinary fields, which is used to measure the strength and direction of the linear relationship between two continuous variables. Pearson correlation coefficient is calculated as follows:

$$r = \frac{n\sum XY - \sum X \sum Y}{\sqrt{\left(n\sum X^2 - \left(\sum X\right)^2\right)\left(n\sum Y^2 - \left(\sum Y\right)^2\right)}}$$

$X$ and $Y$ are two variable observations; $n$ is the number of samples; $\sum XY$ is the sum of the product of all $X$ and $Y$ values; $\sum X$ and $\sum Y$ are the sum of the $X$ and $Y$ values, respectively; $\sum X^2$ and $\sum Y^2$ are the sum of squares of the $X$ and $Y$ values, respectively. The Pearson correlation coefficient ranges from $-1$ to $+1$. $r = +1$ means completely positive linear correlation; $r = -1$ means completely negative linear correlation; $r = 0$ means there is no linear correlation (*Das, Sarkar & Kanungo, 2022*; *Liu et al., 2022*; *Schober, Boer & Schwarte, 2018*).

In this article, the Pearson correlation coefficient is utilized to analyze the correlation between the mean resistance of NPS pollution and the monthly mean value of water quality

indexes at different scales. The purpose is to explore at which scale NPS pollution has the most substantial impact on water quality and which water quality indicator causes the most severe pollution to water quality. This is helpful for determining the scale at which NPS pollution has the most significant influence on water quality, thereby facilitating the formulation and implementation of more effective pollution control measures.

## Determination of "source" landscape type

Landscape types have different ecological functions, with "source" landscapes promoting NPS pollution and having high pollution risk, while "sink" landscapes slow NPS pollution and have low pollution risk. The risk of NPS pollution depends on the intensity of the "source–sink" landscape effect, with a strong source effect leading to high risk and a strong sink effect leading to low risk. Cultivated land and construction land produce much surface runoff and soil erosion, playing a positive role in NPS pollution and showing a strong source effect (*Wang et al., 2017*; *Zhou et al., 2022*). Agricultural land is the primary NPS source (*Shen et al., 2014*; *Zou et al., 2020*). Considering the landscape type distribution and local economy in the Three Lakes Basin, which is dominated by rice planting and industries such as chemicals, building materials, food processing, and aquatic products, paddy fields and construction land landscape patches were determined as the source landscape type of the central Yunnan Plateau lake basin.

## Identification of the "source–sink" risk pattern of NPS pollution based on the MCR model

The identification of NPS pollution risk involves three parts: constructing an NPS pollution resistance base surface evaluation index system, building an NPS pollution resistance base surface, and classifying NPS pollution source–sink risk patterns.

Firstly, NPS pollution is a continuous dynamic process involving rainfall-runoff, soil erosion, sediment transport, and pollutant migration and transformation (*Ouyang et al., 2018*). These processes in the central Yunnan Plateau lake basin are controlled by "source–sink" landscape external natural environment factors, including topography, geomorphology, land use types, and vegetation. Five indexes were selected as key factors of landscape NPS pollution of paddy fields and construction land: relative elevation, relative slope, surface roughness, vegetation coverage, and land use type. The discrete grid values of these factors were linearly normalized and classified into grades 1–5, assigned from low to high as 1, 3, 5, 7, and 9, respectively. The expert scoring method was used to assign weights to each factor, and spatial superposition was performed to obtain the resistance base surface affecting the basin's NPS pollution process (Table 4).

Secondly, the NPS pollution resistance base surface was constructed using the MCR model, superimposing landscape elements and external factors affecting NPS pollution to form the vertical resistance base surface (R) of NPS pollution diffusion. The spatial distance factor (D) was integrated into the MCR model to obtain the accumulated resistance value of the NPS pollution process. The smaller the MCR value, the greater the risk of NPS pollution, and vice versa. The cost-distance module in ArcGIS10.5 was used to generate the NPS pollution process resistance surface and construct the NPS pollution risk resistance pattern of the lake basin in the middle Yunnan Plateau.

**Table 4  Evaluation index of surface resistance.**

| Evaluation index of surface resistance | Index significace |
|---|---|
| Relative elevation (0.20) | The index indicates that each landscape unit is away from the relative elevation of the plateau lake, reflecting the gravity distribution of the NPS pollution process from the source landscape to the plateau lake. The greater the relative elevation is, the greater the gravity effect is, which accelerates the transmission and migration of NPS pollution, and the risk of NPS pollution may be higher. |
| Relative slope (0.17) | The index represents the relative slope of each landscape unit distance to the plateau lake, and represents the acceleration factor of surface erosion runoff dynamics. The greater the slope, the greater the risk of NPS pollution. Slope data were extracted from DEM data of plateau lake basins using ArcGIS 10.5. |
| Surface roughness (0.13) | Using DEM data of plateau lake basin, the surface roughness was calculated by ArcGIS software. Formula: $M = 1/\cos(\partial \times \pi/180)$, where $M$ is the surface roughness and $\partial$ is the slope (°). The rougher the surface, the greater the resistance value, the smaller the risk of NPS pollution. |
| Vegetation coverage (0.24) | Vegetation coverage reflects the underlying surface condition that hinders NPS pollution. The better the vegetation coverage is, the greater the blocking absorption ratio of NPS pollutants passing through, so the regional resistance value of high vegetation coverage is larger. |
| Land use (0.26) | Land use types can distinguish the types and sources of NPS pollution. Construction land, paddy field and dry land are conducive to pollutant output, and woodland and grassland play an effective role in intercepting NPS pollution sources. |

Thirdly, NPS pollution source–sink risk patterns were classified using the surface value of NPS pollution resistance of the paddy field source landscape (PFSL) and construction land source landscape (CLSL) in the lake basin as an index to reflect the risk of source–sink. This index reflects the accessibility of NPS pollutants from the source landscape to the final collection. According to the final resistance surface value grid layer, the natural breakpoint method was used to divide the source–sink risk level of the NPS pollution process affecting the PFSL and CLSL in the basin into five levels: extremely high risk, high risk, medium risk, low risk, and extremely low risk areas. The higher the level, the greater the risk of NPS pollution.

## Multi-scale division of watersheds

In this study, the spatial scales included the sub-basin, riparian buffer, and lakeshore buffer scales. For the sub-basin division, combined with the terrain characteristics of the basin, based on the DEM data of 30m grid spacing and the main rivers into the lake in the study zone, the ArcSWAT tool was used to divide the sub-basin. Finally, the Fuxian Lake Basin was divided into nine sub-basins of different sizes, namely, the Daicun River, Dongda River, Gehe River, Jianshan River, Liangwang River, Luju River, Fucheng River, Niumo River, and Shanchong River Basins. The Xingyun Lake Basin was divided into eight sub-basins of different sizes, namely, the Dajie River, Dalongtan River, Dazhuang River,

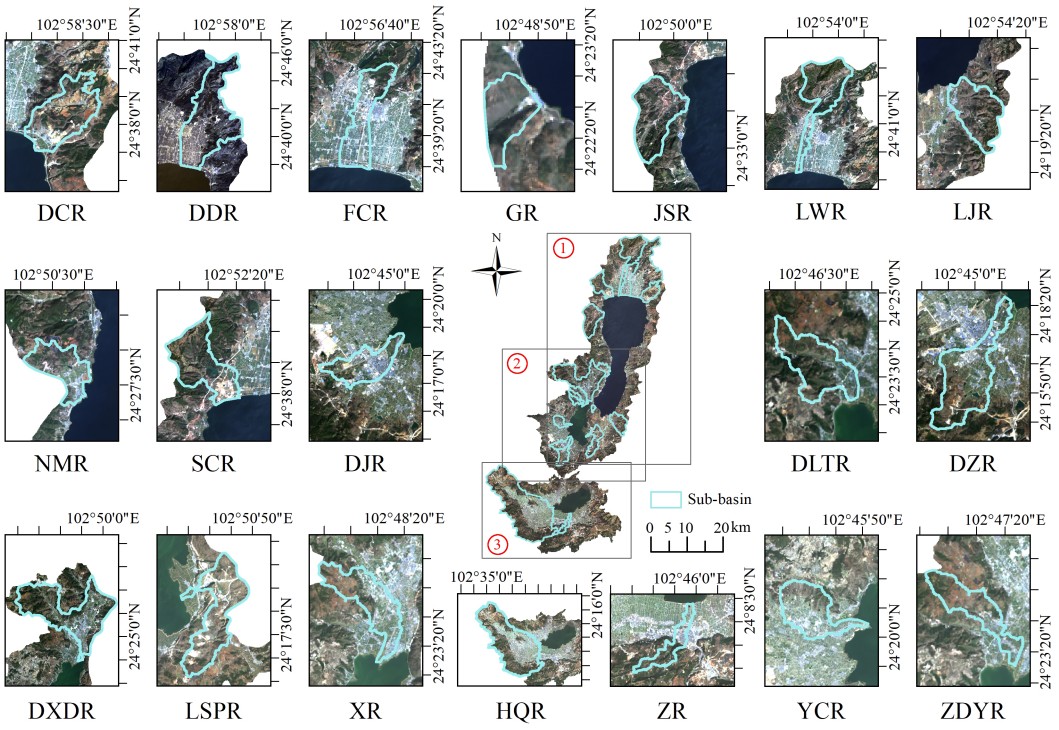

**Figure 2  Overview of sub basins of Fuxian Lake, Xingyun Lake, and Qilu Lake.** Marks ① to ③ indicate Fuxian Lake, Xingyun Lake, Qilu Lake,respectively. The location of nine major sub-basins in Fuxian Lake, including Daicun River (DCR), Dongda River (DDR), Fucheng River (FCR), Ge River (GR), Jianshan River (JSR), Liangwang River (LWR), Lujv River (LJR), Niumo River (NMR), Shanchong River (SCR); The location of eight major sub-basins in Xingyun Lake, including Dajie River (DJR), Dalongtan River (DLTR), Dazhuang River (DZR), Dongxida River (DXDR), Luosipu River (LSPR), Xue River (XR), Yu-cun River (YCR), Zhoudeying River (ZDYR); the location of two major sub-basins in Qilu Lake, including Hongqi River (HQR), Zhong River (ZR).

Dongxi River, Luosipu River, Xue River, Yucun River, and Zhoudeying River. The Qilu Lake Basin was divided into two sub-basins of different sizes, namely, the Hongqi River and Zhonghe River (Fig. 2). The scale of the riparian and lakeshore buffer zone was the buffer zone generated based on the main rivers flowing into the lake and the lake shoreline. According to previous research findings and the current state of the basin (*Li et al., 2019*; *Peng & Li, 2021*), riparian buffer zones of 100 m, 300 m, 500 m, 700 m, and 1,000 m and lakeshore buffer zones of 110 m were used.

## RESULTS

### Distribution characteristics of source landscape patches and NPS pollution risk resistance in the Three Lakes Basin

The patches of PFSL in the Three Lakes Basin were mainly distributed in flat areas, clustered around the main rivers entering the lake, especially along the Shanchong, Liangwang, Fuorange, East, Niumo, Xuehe, Zhoudeying, Dalongtan, Yuecun, Dajie, Dazhuang, and the middle and lower reaches of the Hongqi River. The patches of CLSL were primarily

**Table 5  Area proportion and area of PFSL and CLSL patches in the Three Lakes Basin.**

| Basin | PFSL | | CLSL | |
|---|---|---|---|---|
| | Area/km² | Area ratio/% | Area/km² | Area ratio/% |
| Fuxian Lake | 66.9978 | 14.62% | 19.2573 | 4.20% |
| Xingyun Lake | 70.9839 | 22.73% | 23.6214 | 7.57% |
| Qilu Lake | 98.4564 | 24.24% | 32.9472 | 8.11% |

scattered in the flat terrain around the lake. The area of PFSL patches was larger than that of CLSL patches, with both showing increasing trends from Fuxian Lake and Xingyun Lake to the Qilu Lake watershed, which had the largest area of PFSL and CLSL patches (Table 5). The wide distribution of PFSL patches indicated that paddy fields were crucial for the local agricultural economy.

The risk distribution of NPS pollution resistance surfaces in the Three Lakes Basin, based on the MCR model, is shown in Fig. 3, where "value" represents the MCR value. The distribution characteristics of the NPS pollution risk resistance of the PFSL and CLSL in the Three Lakes Basin revealed that the closer the distance to the main river trunk road entering the lake, the smaller the MCR value, and the more likely the NPS pollutants are to migrate to the adjacent river and cause water pollution. Conversely, at larger distances from the main river, the greater the resistance face value, the lower the accessibility of NPS pollution to the river, and the smaller the risk of water being polluted by NPS pollution. The change trend of the MCR value showed a decreasing trend from the steep mountainous area to the lake shoreline, with the minimum resistance value near the lake shoreline, indicating that the risk of NPS pollution in mountain areas was less than that in flat terrain areas, and the river water quality near lake shores was most affected by NPS pollution. The distribution area of low resistance values of PFSL was larger than that of CLSL, and the spatial distribution range of paddy fields was wider than that of construction land (Table 5). According to the MCR model theory, the closer other landscape patches are to the source landscape, the smaller their resistance value, the stronger the role of NPS pollution sources, and the more likely the pollutants are to migrate into the river and cause pollution.

## Distribution characteristics of "source–sink" risk levels of NPS pollution in the Three Lakes Basin

The risk zones of NPS pollution "sources–sinks" of the PFSL and CLSL patches in the Three Lakes Basin were divided into five levels using the natural breakpoint method: extremely high risk, high risk, medium risk, low risk, and extremely low risk zones. The higher the risk level, the greater the risk of NPS pollution. The NPS pollution in the Three Lakes Basin under the influence of the PFSL and CLSL was in high risk zones, with the paddy field "source" effect being stronger than that of the construction land source. The distribution ranges of the extremely high and high risk levels were larger for PFSL than for CLSL (Fig. 4). In the Three Lakes Basin, the proportion of extremely high risk and high risk zones was greater than 60% for both PFSL and CLSL (Fig. 5). The extremely high risk zones were continuously concentrated along the lake shoreline and the main rivers flowing

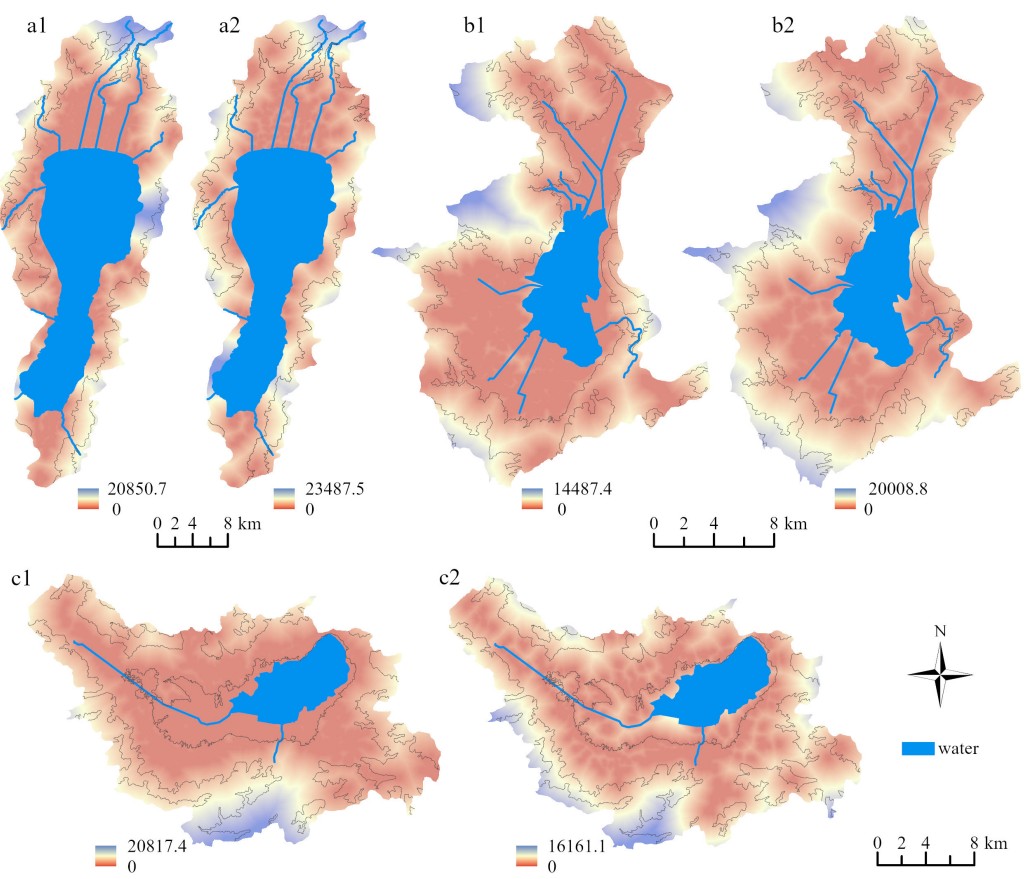

**Figure 3   Resistance values of PFSL and CLSL in three lake basins.** (A1) Paddy field and (A2) construction land in Fuxian Lake Basin. (B1) paddy field and (B2) construction land in Xingyun Lake Basin. (C1) paddy field and (C2) construction land in Qilu Lake Basin.

into the lake. The mountainous terrain undulation zone was a low risk zone with high resistance values due to the steep terrain aiding in the rapid diffusion and migration of pollutants, but the high vegetation coverage intercepting the migration and transmission of pollutants. In the plain zone near the lakeshore, towns are clustered, and PFSL is densely distributed, resulting in a high risk of NPS pollution due to the increased likelihood of pollutants entering water bodies through rainfall and surface runoff.

## Multi-scale correlation analysis between NPS pollution risk resistance and water quality in the Three Lakes Basin

The mean value of the NPS pollution risk resistance surface of the PFSL and CLSL at different scales was negatively correlated with the monthly mean value of water quality indexes of inflow rivers (Figs. 6A, 6B). Except for the lakeshore scale, the mean resistance surface of sub-basins and riparian buffer zones (100 m, 300 m, 500 m, 700 m, and 1,000 m) was significantly correlated with the monthly mean values of $NH_3$-N, TP, and $COD_{Cr}$ for PFSL, and $NH_3$-N, TP, TN, $BOD_5$, and $COD_{Cr}$ for CLSL. The 1,000 m riparian buffer scale had the strongest correlation with most water quality indexes, especially $NH_3$-N. The

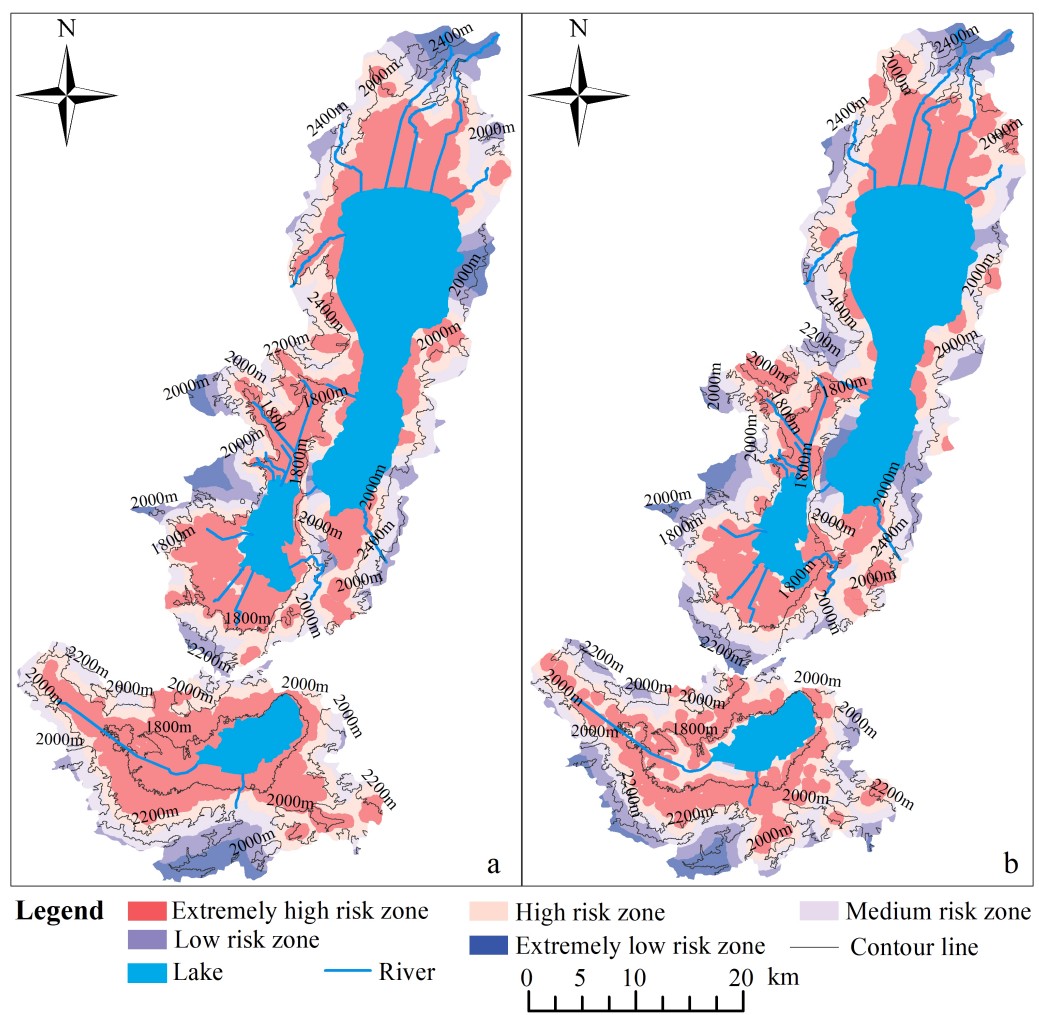

**Figure 4  Distribution of risk levels for NPS pollution.** (A) PFSL, (B) CLSL.

correlation between the mean resistance of the CLSL and the monthly mean water quality indexes was significantly higher than that of the PFSL, showing a significant negative correlation. $NH_3$-N was an important water quality indicator affecting river water quality, particularly at the 1,000 m riparian buffer scale.

In the correlation analysis of the PFSL and CLSL, the correlation value of the riparian buffer scale was higher than that of the sub-basin scale. With the increase in riparian buffer scale distance from 100 m to 1,000 m, the correlation value increased, except for $COD_{Mn}$. The mean value of the resistance surface at the lakeshore scale had no obvious correlation with the various water quality indexes, possibly due to the small number of PFSL and CLSL patches and the limited zone size at the 110 m scale.

Paddy fields and construction land in the Three Lakes Basin were the main sources of NPS pollution, with water bodies at the seven scales showing different degrees of NPS pollution risks. The multi-scale analysis revealed that the 1,000 m riparian buffer scale was

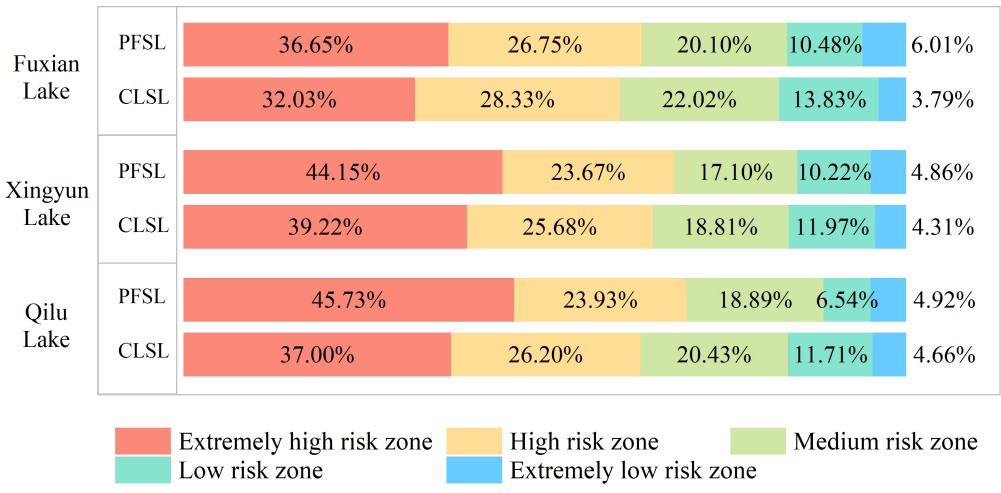

**Figure 5** **The proportion of NPS pollution risk levels for PFSL and CLSL in Three Lake Basins.**

the most sensitive to the impact of NPS pollution on water quality, as it had the strongest correlation between NPS pollution risk resistance and water quality indexes.

## DISCUSSION

### Differences in the risk distribution of NPS pollution in the Three Lakes Basin

The risk of NPS pollution in the Three Lakes Basin was high, with paddy fields and construction land being the primary sources. Near the main rivers flowing into the lakes and in flat terrain zones, the MCR value was small, and the risk of NPS pollution was high, consistent with previous studies (*Kong et al., 2018*; *Wang et al., 2018*). The maximum resistance surface value of the PFSL in Fuxian Lake and Xingyun Lake was smaller than that of the CLSL, while in Qilu Lake, it was larger.

The spatial range of the source landscape distribution usually indicates the severity of NPS pollution, with a larger range corresponding to a smaller resistance value (*Ouyang et al., 2010*). Although the paddy field zone in the Qilu Lake Basin was larger than the construction land zone, the maximum resistance surface value of the PFSL was larger than that of the CLSL (Fig. 7). This result may be due to the flat terrain along the lake shoreline and near the main rivers, where the interception effect of source pollution is weak, and the resistance value is small. Additionally, the close distance and strong landscape connectivity between construction land patches on the southern edge of the Qilu Lake Basin may contribute to this result, as better connectivity leads to a smaller resistance value and greater risk of NPS pollution (*Wang & Pan, 2019*).

The extremely high and high risk zones in the Three Lakes Basin were significantly larger than the low and extremely low risk zones, which were mainly distributed in rugged terrain. The size of the extremely high and high risk zones was mainly affected by the type of source landscape, distribution space, aggregation, and spatial distance (*Wang et al., 2019b*). The PFSL and CLSL in the basin were large, continuous, and located in flat

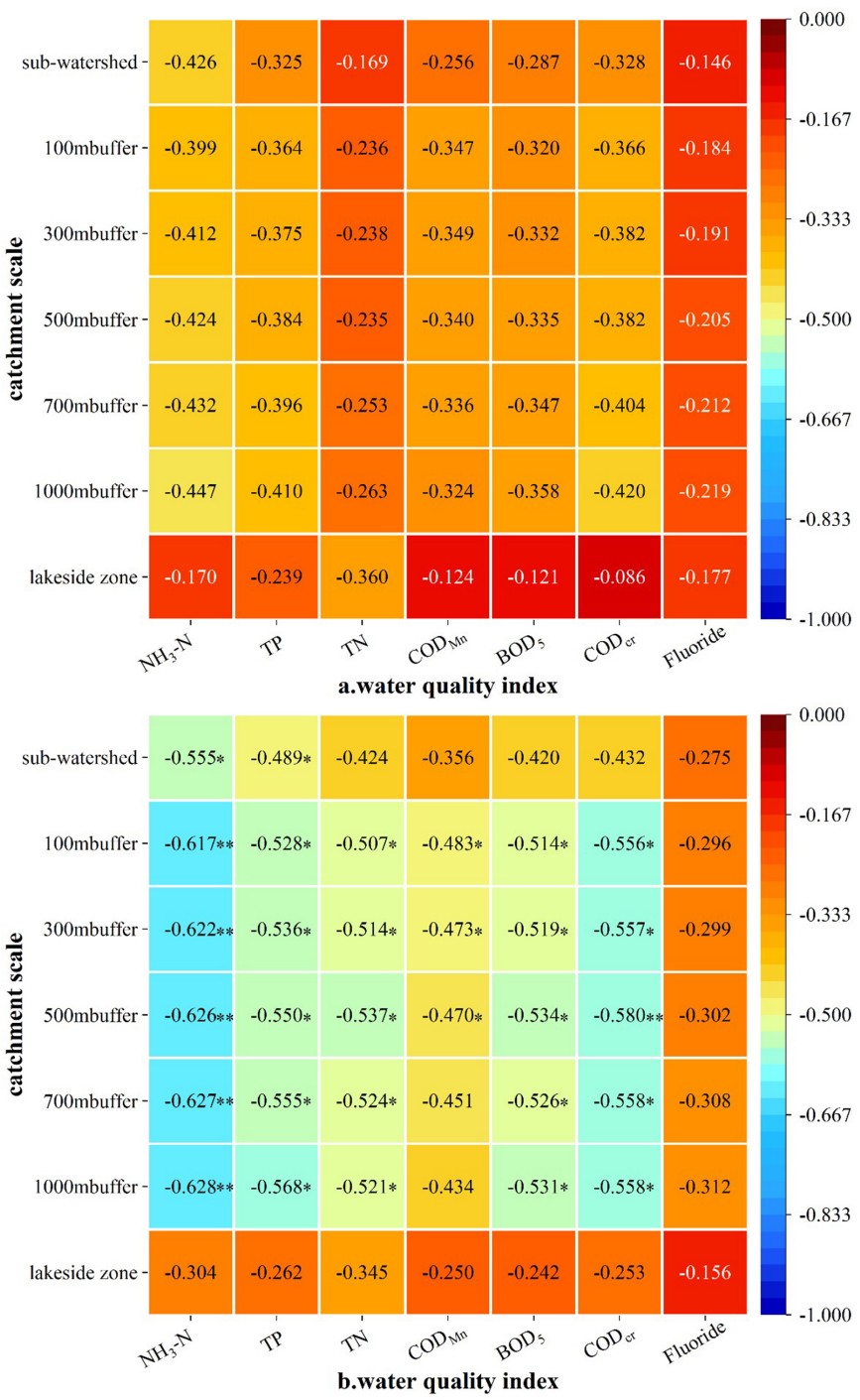

**Figure 6  Multi scale correlation analysis of mean NPS pollution risk resistance and monthly mean water quality index in Three Lake Basins.** Notes ** indicates Sig. is a significant correlation at the 0.01 level (bilateral); * indicates Sig. is a significant correlation at the 0.05 level (bilateral). (A) The result of multi-scale correlation analysis of PFSL. (B) The result of multi-scale correlation analysis of CLSL.

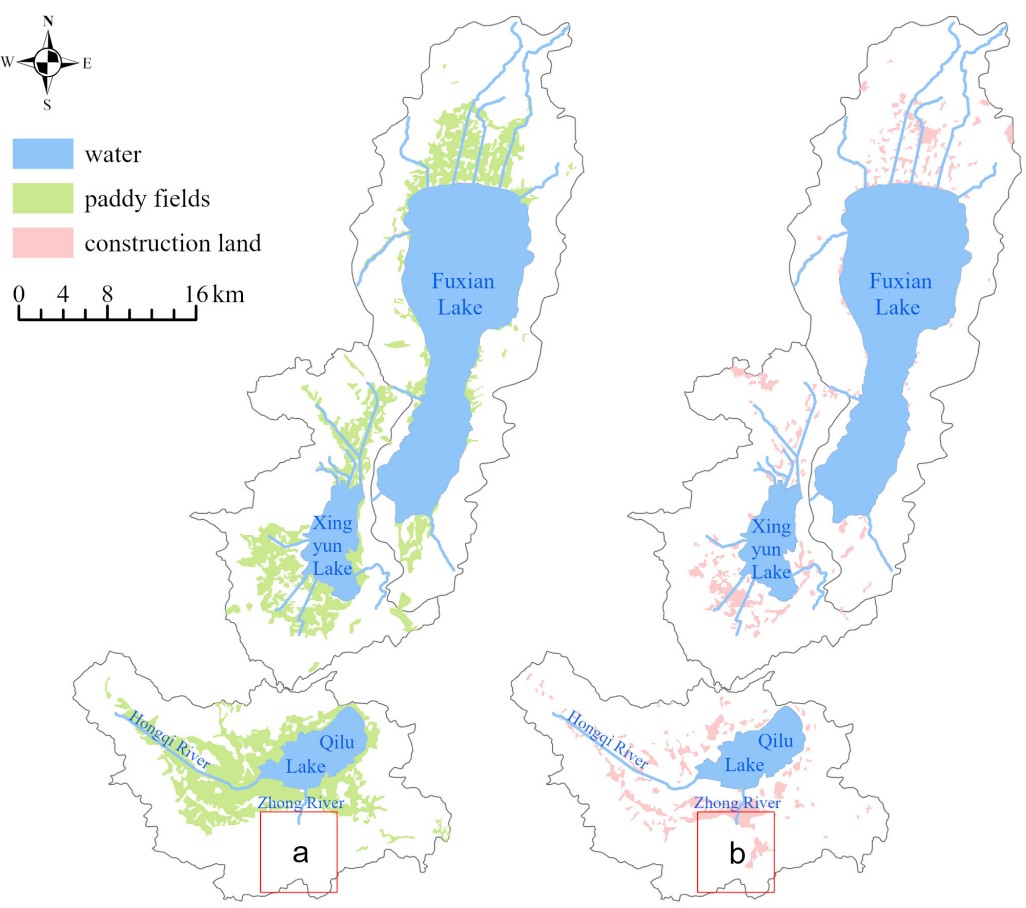

**Figure 7 Patch distribution of PFSL and CLSL in Three Lake Basins.** (A) The distribution of paddy field in Qilu Lake; (B) The distribution of construction land in Qilu Lake.

zones near the main rivers, with weak pollutant interception ability. The clustered and distributed landscape patches near the river and lake shoreline were more likely to produce and transport pollutants to rivers, causing water pollution. Future studies should consider the effects of various spatial distribution characteristics of source landscape patches on NPS pollution.

## Relationship between NPS pollution risk resistance value and water quality

The correlation analysis showed that the negative correlation between the mean value of source pollution resistance and water quality indexes at different scales was relatively high, indicating that water quality was seriously affected by NPS pollution from PFSL and CLSL. At the 1,000 m riparian buffer scale, $NH_3$-N had the highest correlation with the mean value of NPS pollution resistance of the two source landscapes, making it the most sensitive index of water quality to NPS pollution. These results were related to the source landscape types, landscape distribution pattern, land use patterns, and topographical conditions of the study zone (*Zhao et al., 2015*; *Bi et al., 2021*).

Comparing the correlation analysis results of paddy fields and construction land revealed that $NH_3$-N, TP, and $COD_{Cr}$ were the most sensitive indexes of water pollution in sub-basins and riparian buffer zones (100 m, 300 m, 500 m, 700 m, and 1,000 m) for PFSL, while $NH_3$-N, TP, TN, $BOD_5$, and $COD_{Cr}$ were the most sensitive indexes for CLSL. The number of sensitive water quality indexes in the CLSL was larger than that in the PFSL at the sub-basin and riparian buffer scales. This difference can be attributed to several factors: (1) The impervious surface of urban construction land increases the concentration of pollutants in surface runoff, indirectly increases runoff, increases the output of NPS pollution, and promotes an increase in nutrients in water quality (*Berland et al., 2017*). Studies have shown that the output of NPS pollution from urban construction can be even higher than that of cropland (*Dunn et al., 2014*; *Shi et al., 2017*). (2) The lack of centralized treatment for domestic waste and sewage from farmers in the Three Lakes Basin aggravates the discharge of nutrients into rivers. (3) China's economic development policy, which prioritized rapid industrial growth and GDP expansion, led to the introduction of enterprises that rely on local land resources for operation and production, resulting in the continuous discharge of deleterious effluents, excessive harmful gases, and floating dust sources into rivers with rainfall runoff, causing serious NPS pollution.

To control NPS pollution in the future, the scale of construction land should be reasonably planned, the zone of the impervious layer should be reduced, the vegetation of the river buffer zone should be protected, and relevant departments should coordinate basin planning and administrative zone planning.

## The scale effect of NPS pollution on water quality

The impact of NPS pollution on water quality varies at different scales, with studies showing varying results. *Dai et al. (2017)* found that the 200 m riverbank buffer zone was key (*Dai et al., 2017*), while Peng Shuangyun (*Peng & Li, 2021*) found the 300 m and 500 m riparian buffer zones in farmland and villages had the most impact. Li Shihua (*Li et al., 2019*) showed that the 100 m riparian buffer zone had a more significant correlation between land use/land cover (LULC) and water quality. This study revealed that NPS pollution had the greatest impact at the 1,000 m riparian buffer scale.

Landscape spatial distribution patterns, influenced by topography, soil, vegetation, and human activities, can aggravate NPS pollution if unreasonable (*Sun, Ma & Fang, 2023*). The "source–sink" landscape distribution greatly influences NPS pollution formation, with cropland and construction land as sources and woodland and grassland as sinks. Table 6 shows that at the sub-basin and riparian buffer scales, patch zones of cropland, forestland, grassland, and construction land increased with spatial scale, except for cropland at the sub-basin scale. In the 1,000 m riparian buffer zone, construction land exceeded the "sink" landscape's ability to purify and absorb pollutants. The lakeshore scale had no significant correlation due to lower landscape distribution zones. PFSL and CLSL promoted NPS pollution but had small distribution zones and generated pollutants. Forestland, grassland, and soil intercepted and purified NPS pollutants, weakening their spread at the sub-basin scale.

**Table 6 Landscape patch area at different types of scales (km²).**

| Area of landscape patche/spatial scale | Cropland | Forestland | Grassland | Construction land |
|---|---|---|---|---|
| 100 m buffer | 0.91 | 6.37 | 2.28 | 16.59 |
| 300 m buffer | 2.00 | 18.13 | 8.70 | 48.30 |
| 500 m buffer | 2.72 | 30.42 | 16.03 | 77.11 |
| 700 m buffer | 3.20 | 42.47 | 23.46 | 105.61 |
| 1,000 m buffer | 3.60 | 60.73 | 35.35 | 145.13 |
| sub-basin | 2.92 | 152.35 | 70.10 | 219.97 |
| lakeside zone | 0.06 | 0.01 | 0.02 | 1.71 |

The 1,000 m riparian buffer zone had the most significant impact on water quality. To reduce NPS pollution risk in extremely high and high risk zones, the "source" effect of paddy fields and construction land should be weakened, and the "sink" effect strengthened (*Ma et al., 2021*; *Wu et al., 2016b*). In the Three Lakes Basin, less frequent use of chemical fertilizers and pesticides, farmland irrigation with sewage, and standardized urban sewage discharge policies are suggested (*Sun et al., 2012*). Riparian vegetation buffer zones, multifunctional ecological fish ponds, and river protection zones can reduce the impact of NPS pollution on water quality (*Mi et al., 2015*; *Zhang et al., 2013*).

Future research should consider time series to understand temporal variations in the relationship between NPS pollution and water quality, incorporate detailed land use classifications and characteristics, and integrate factors like climate, soil properties, and management practices to develop comprehensive strategies for mitigating NPS pollution and protecting water quality.

## CONCLUSION

Compared to point source pollution, nonpoint source (NPS) pollution presents greater challenges in terms of detection and control due to its widespread occurrence, leading to the environmental introduction of pollutants through diffusion. Based on the geographical location and related characteristics of the study area, this study selected PFSL and CLSL patches, which are relatively dispersed pollution sources and difficult to quantify using traditional methods. The MCR model is specifically used for assessing the risk of NPS pollution by incorporating various resistance factors into the analysis to comprehensively evaluate risk distribution. This article utilizes the MCR model to analyze the impact of NPS pollution on water quality at different scales, revealing the risk distribution of NPS pollution in the Three Lakes Basin and its impact on water quality. The results indicate that PFSL and CLSL are mainly located in the flat areas surrounding lakes, presenting an overall concentrated distribution. They are the primary reasons for giving rise to the risk of NPS pollution. This article conducts a comparative analysis of NPS pollution between PFSL and CLSL. The results show that the distribution area of PFSL is larger than that of CLSL, and the NPS pollution risk of PFSL is greater than that of CLSL. This result effectively supplements relevant content regarding specific farmland types' impact on regional ecological environments while providing valuable insights for further exploration

into PFSL's impact on nutrient input into surrounding water bodies within the region. Over 60% of the basin exhibits high or extremely high risk levels, primarily situated in flat areas around lakes and along main rivers flowing into them. This article employs multiple scales to analyze the impact of NPS pollution on water quality. The results show that there are significant scale differences in the impact of NPS pollution on the water quality of rivers entering the lake. In particular, the impact is greatest at the 1,000 m riparian buffer scale. These results offer an accurate basis for selecting future research scales while aiding in formulating targeted measures for effective pollution prevention and control based on distinct scale correlation characteristics; such as strengthening monitoring efforts within 1,000 m riparian buffer zones.

## ACKNOWLEDGEMENTS

The authors are very grateful to the editor and anonymous reviewers for their valuable comments and helpful suggestions.

### Funding

The National Natural Science Foundation of China (42261073, 41971369, 41561086, 42201037), the Yunnan Province Reserve Talent Program for Young and Middle-aged Academic and Technical Leaders (202305AC160083, 202205AC160014), and the Yunnan Provincial Basic Research Project (202401AT070103, 202201AS070024, 202001AS070032). The funders had no role in study design, data collection and analysis, decision to publish, or preparation of the manuscript.

### Grant Disclosures

The following grant information was disclosed by the authors:
The National Natural Science Foundation of China: 42261073, 41971369, 41561086, 42201037.
Yunnan Province Reserve Talent Program for Young and Middle-aged Academic and Technical Leaders: 202305AC160083, 202205AC160014.
Yunnan Provincial Basic Research Project: 202401AT070103, 202201AS070024, 202001AS070032.

### Competing Interests

The authors declare there are no competing interests.

### Author Contributions

- Li Fu conceived and designed the experiments, performed the experiments, analyzed the data, prepared figures and/or tables, authored or reviewed drafts of the article, and approved the final draft.
- Xiaoliang Ma conceived and designed the experiments, performed the experiments, analyzed the data, prepared figures and/or tables, authored or reviewed drafts of the article, and approved the final draft.

- Shuangyun Peng conceived and designed the experiments, performed the experiments, analyzed the data, prepared figures and/or tables, authored or reviewed drafts of the article, and approved the final draft.
- Luping Gong conceived and designed the experiments, analyzed the data, prepared figures and/or tables, authored or reviewed drafts of the article, and approved the final draft.
- Rui Zhang conceived and designed the experiments, analyzed the data, prepared figures and/or tables, authored or reviewed drafts of the article, and approved the final draft.
- Bangmei Huang conceived and designed the experiments, analyzed the data, prepared figures and/or tables, authored or reviewed drafts of the article, and approved the final draft.

## Data Availability

The raw measurements are available in the Supplementary File.

## Supplemental Information

Supplemental information for this article can be found online at http://dx.doi.org/10.7717/peerj.18247#supplemental-information.

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
