# Peer review of "Exploring the scale effect of nonpoint source pollution risk on water quality in Lake Basins of Central Yunnan Plateau using the Minimum Cumulative Resistance model"

_PeerJ, doi:10.7717/peerj.18247_

## Round 0.1 · original submission · Major Revisions

Scale effects have widely (and best) studied in landscape ecology, but you seem unaware of the key literature in this area. I suggest that you read some of the papers on scale effects and multiscale analysis, so you can cast your results in a broader and proper context. Here some references for you to get started:

Sun, X., Ma, Q., Fang, G., 2023. Spatial scaling of land use/land cover and ecosystem services across urban hierarchical levels: patterns and relationships. Landsc. Ecol. 38, 753-777.

Wu, J., 2004. Effects of changing scale on landscape pattern analysis: Scaling relations. Landsc. Ecol. 19, 125-138.

Wu, J., 2007. Scale and scaling: A cross-disciplinary perspective, in: Wu, J., Hobbs, R. (Eds.), Key Topics in Landscape Ecology. Cambridge University Press, Cambridge, UK, pp. 115-142.

·

Basic reporting

The manuscript is generally well-written and uses professional English. However, some sentences are too short and informal for academic writing. For example, see lines 54 and 77.

The introduction provides solid context and justifies the necessity of this study. However, more recent references (within the last 3-5 years) should be included to ensure the literature review is up-to-date. For instance, in lines 75-76, "the MCR model has been used to assess NPS pollution risk in various regions in China," the only references are from 2016 and 2018.

Figures and Tables need to be revised:

Figure titles should be placed at the bottom of each figure.
Figures 1-4 can be combined into two figures for better clarity (for your consideration).
Figure 5 could be moved to the supplementary section (for your consideration).
Figure 7 should be replaced with a bar chart, as a line chart is not suitable for representing non-continuous data.
The use of colors in some figures is confusing (e.g., Figure 8). Typically, red indicates high risk (danger), and green indicates low risk (safe). Reversing these colors may cause confusion.
All tables and figures need significant modifications to ensure they are well-labeled and thoroughly described in the text.

Experimental design

The research question is well-defined, and this study is relevant within the scope of this journal.

The Methods section needs clarification:
What is the Minimum Cumulative Resistance (MCR) model, and why did the authors choose it over other methods? A comparison with other methods would be helpful.

The terms "PFSL and CLSL" (line 181) should be fully spelled out the first time they appear in the text.

Consider reorganizing the "Materials and Methods" section to improve flow and readability.

Validity of the findings

The conclusions need to highlight the true and innovative findings of this research. The statement "The results showed that the Three Lakes Basin is facing serious NPS pollution risks" (line 372) is too general and cannot be considered a robust conclusion. Similarly, the statement "Affected by various natural and human factors,... and random locations" (lines 377-379) is more of a common observation and not a specific conclusion derived from this research. The authors should focus on the unique contributions of their study, such as specific insights on scale effects of NPS pollution, novel applications of the MCR model, or new findings regarding pollution sources and impacts in the Three Lakes Basin.

Additional comments

The manuscript needs significant reshaping, including major modifications in the experimental design, data presentation (tables/figures), and final conclusion. In its current form, it is not suitable for publication as a research outcome.

Consider explaining 1-2 key findings in detail and demonstrating their effects or significance by comparing them with previous research or highlighting their potential value for future studies.

Reviewer 2 ·

Basic reporting

Many missing details, undefined abbreviations, and many necessary figures

Experimental design

Details are not enough

Validity of the findings

Not relevant

Additional comments

The paper is interesting but as of now has many gaps, such that a reader will find it extremely difficult to understand. Please ask someone other than the authors could understand (it is natural the authors to understand even with many gaps in methodology etc).
The abstract is not representative. Has allocated more for the introduction, but not for results and discussion.
L47: Briefly explain, perhaps citing from the same paper, how and why it was stated that NPS is the most responsible (over-point source)
Table 3: State the units.
L72: MCR, use the full term before using the abbreviation. The introduction should have more explanation and use of MCR, as it is one of the main components of the study. Also, the methodology needs to have more explanations/content related to MCR.
List objectives or hypotheses at the end of the introduction.
PFSL and CLSL not defined?
The paper attributes pollution to NPS, but I do not see any rationale for excluding point source pollution. Isn't it possible to show the contribution in your study area as a %? Or simply without such attribution investigate the impact of NPS on WQ
The paper is not reader-friendly, and difficult to understand. This is because there are many missing explanations as and when needed (e.g. MCR in the introduction); undefined abbreviations, and some extremely poor figures (e.g., Fig 4, I can’t figure out what the subbasins, and where are those rivers referred in the caption); Fig 6: Construction land? What the legend (color code) stands for?
# There are many unnecessary figures/sub-figures. E.g., Fig 2, the left side large figure enough. Instead the authors have shown separate figures for every sub basin. Is it needed? Same issue with Figure 3
Is Figure 5 needed, marking the buffers.
Table 3: Are this the average of monthly values?
Fig 9: Multiscale correlation analysis; have you sufficiently explained this under methodology? Have you used a software? Is this use Pearson’s correlation or something similar?
Somewhere mention this is purely a spatial study.
Thank you

---

## Round 0.2 · accepted · Accept

Thank you for revising your manuscript. I'm happy to accept it for publication in PeerJ.

·

Basic reporting

The revised manuscript has effectively addressed most of the previous reviewers' comments. The authors have made detailed modifications compared to the first version, significantly improving the quality and making it ready for publication.

Experimental design

The experimental design is much clearer than in the previous version I reviewed. The figures and tables have also been improved, making them more suitable for public reading and review.

Validity of the findings

The authors focused on 1-2 key findings and made comparisons with previous research, enhancing the validity of the experimental results.

Both the abstract and conclusion are more precise and aligned with the core experimental design and the results obtained in this research.

Additional comments

The revised version is in good shape for publication after carefully checking the language, spelling, and other details.